# Bleach-Induced Chemical Sinusitis and Orbital Cellulitis Following Root Canal Treatment

**Terese Huiying Low \*, Jun Jie Seah** **, Somasundaram Subramaniam, Vijayaraj Thirunavukarasu and Chew Lip Ng**

Department of Otolaryngology—Head and Neck Surgery, Ng Teng Fong General Hospital, National University Health System, Singapore 609606, Singapore

\* Correspondence: terese_low@nuhs.edu.sg; Tel.: +65-6908-2222

**Abstract:** The authors describe an unusual case of chemical sinusitis and orbital cellulitis secondary to a sodium hypochlorite accident in a patient who had just undergone root canal treatment. The patient presented with acute, progressive symptoms of unilateral maxillary sinusitis, facial cellulitis and orbital cellulitis which began hours after root canal treatment on the ipsilateral side. He was admitted to hospital under the care of the Otorhinolaryngology team and reviewed regularly by the Ophthalmologists. He underwent Endoscopic Sinus Surgery during his hospital stay. The intra-operative findings revealed necrotic sinus mucosa and slough within the involved maxillary sinus, which were suggestive of chemical burn injury induced by the highly alkaline sodium hypochlorite solution used during root canal treatment. He was treated postoperatively with regular nasal toilet, culture-directed antibiotics and topical ocular pressure-lowering eyedrops. He displayed a slow recovery with eventually no orbital sequelae, but experienced persistent cheek numbness three months post-injury. Severe chemical sinusitis with orbital cellulitis secondary to sodium hypochlorite accident is a rare complication of root canal treatment, with potentially severe consequences. It can present with symptoms similar to complicated acute bacterial sinusitis. Otorhinolaryngologists and dental surgeons should maintain a high index of suspicion when managing a patient post-root canal treatment with symptoms of unilateral sinusitis, facial cellulitis, orbital cellulitis and even airway compromise. This would allow prompt intervention before sight or life-threatening complications set in.

**Keywords:** chemical sinusitis; bleach; hypochlorite; dental; root canal; orbital cellulitis

## 1. Introduction

In this case report, we describe our experience with a patient diagnosed with a rare case of left-sided chemical maxillary sinusitis, complicated by ipsilateral orbital cellulitis secondary to hypochlorite accident during a recent root canal treatment. We use the term "chemical sinusitis" to refer to an inflammatory process of the paranasal sinus tissue due to an offending chemical agent that is toxic to the mucosal lining.

## 2. Case Report

A 64-year-old Chinese Male presented with progressive left-sided facial swelling over five days. This started after routine root canal treatment on his left third maxillary molar. 5% sodium hypochlorite solution was used as the irrigant. He developed ipsilateral cheek numbness and eye pain the day after the procedure. His symptoms were not responsive to oral antibiotics and analgesia prescribed by his Dentist.

At presentation, our patient was afebrile. There was left-sided proptosis, chemosis, and periorbital edema, with marked facial swelling and cheek numbness. Extraocular movements were restricted. Pupillary reflexes were normal. Nasoendoscopy showed mucosal congestion within the left middle meatus.

Leukocyte count and C-reactive protein were elevated. A Computed Tomography (CT) scan of the paranasal sinuses (Figure 1) showed opacification with gas pockets within the left maxillary sinus with extensive bony erosion of the sinus walls.

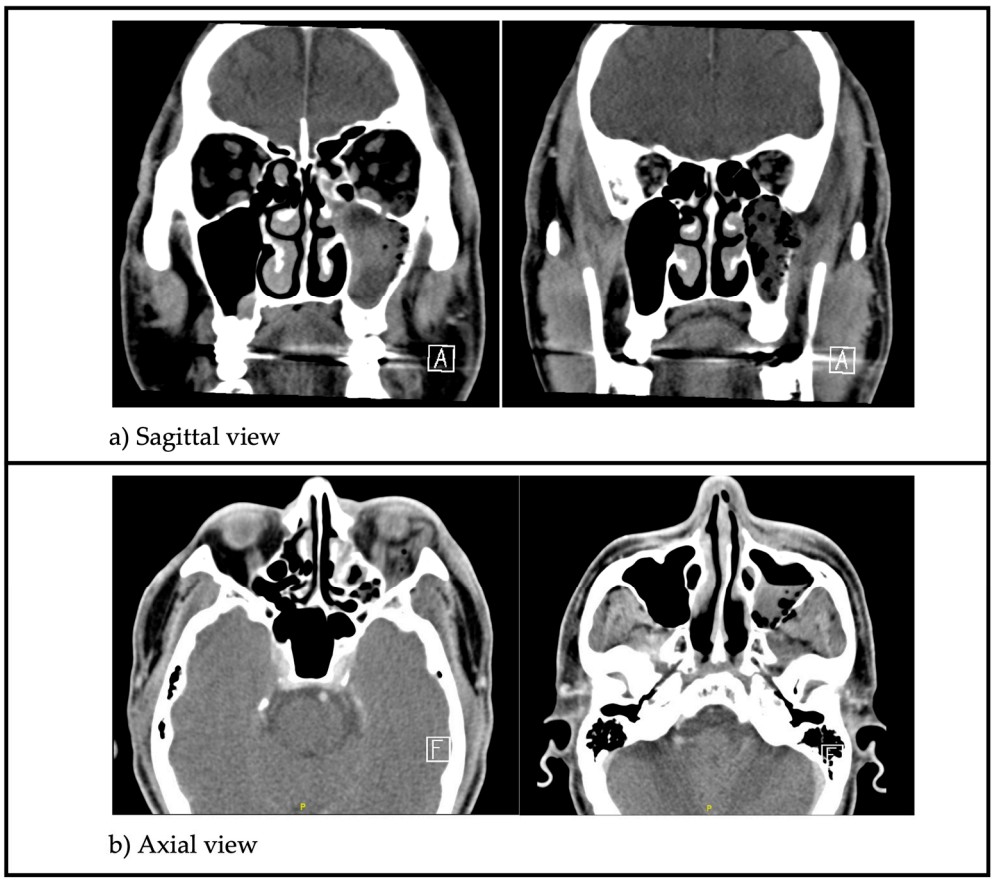

a) Sagittal view

b) Axial view

**Figure 1.** (**a**) Sagittal CT images showing opacification of the left maxillary sinus with gas pockets within it. There is significant bony erosion of the sinus walls; (**b**) Axial CT images highlighting soft tissue gas and inflammation extending beyond the left maxillary sinus to involve the orbit superiorly, infratemporal fossa laterally and subcutaneous tissue of the cheek anteriorly.

Soft tissue inflammation extended beyond the sinus to involve the orbit superiorly (Figure 2), infratemporal fossa laterally and subcutaneous tissue of the cheek anteriorly. The roots of the second and third maxillary molars were seen protruding into the maxillary sinus (Figure 3).

The collective impression of the Otorhinolaryngology, Ophthalmology, and Dental teams was that of left-sided acute chemical-induced maxillary sinusitis, facial and orbital cellulitis secondary to sodium hypochlorite accident. This was given the history of rapid onset of symptoms following root canal treatment, and physical examination findings.

The patient underwent Left Endoscopic Sinus Surgery (Middle Meatal Antrostomy). Intra-operatively, the left maxillary sinus was filled with turbid brown fluid and dusky mucosa (Figure 4). There was no frank pus. The sinus was thoroughly irrigated with normal saline. Histology of the unhealthy mucosa revealed fibrotic subepithelial stroma and acute-on-chronic inflammatory cellular infiltrates with foci of dystrophic calcification. Tissue cultures grew *Streptococcus constellatus* and *Citrobacter koseri*. There were no fungal elements seen on fungal smear or culture.

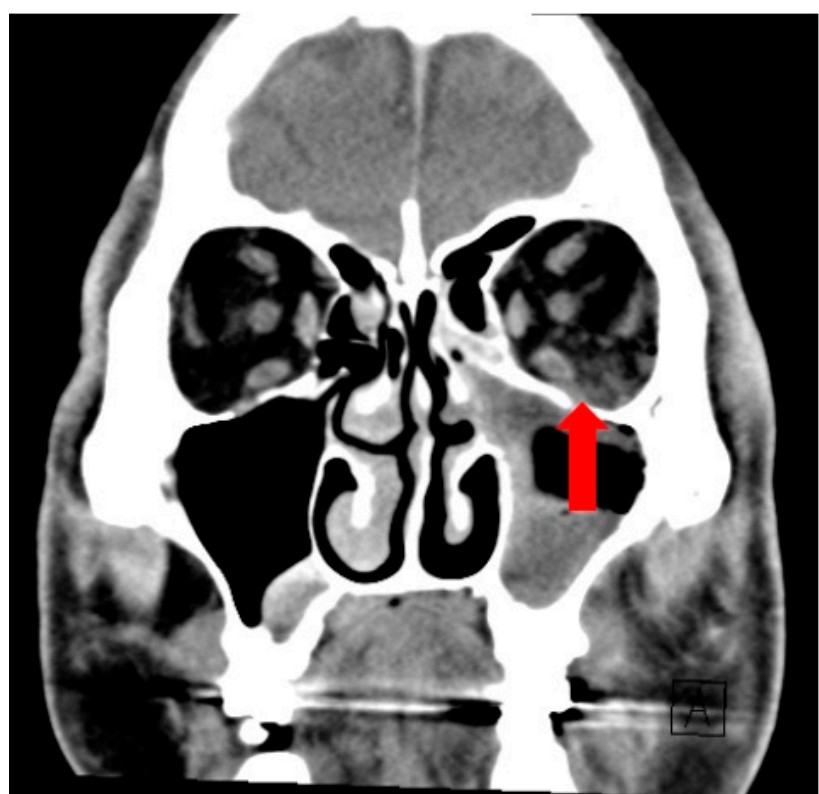

**Figure 2.** Coronal CT image showing fat stranding in the left extra-conal orbital fat (red arrow).

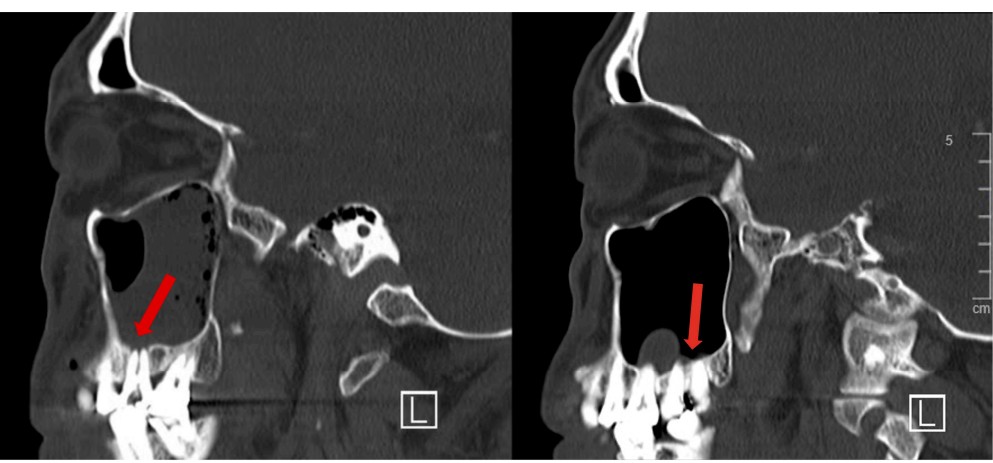

**Figure 3.** Sagittal CT images from the same series showing projection of maxillary tooth roots into the maxillary sinus, with dehiscent overlying bony covering (red arrows).

Culture-directed antibiotics comprising intravenous Cefepime and Metronidazole were commenced. Improvement of symptoms was observed within 24 h post-surgery and was most marked in the first 72 h. Antibiotics were then changed to oral Amoxicillin-Clavulanic acid and the patient completed a total of two weeks of antibiotics. The patient was also prescribed oral steroids and sinus saline irrigation. He reported a burning sensation in his nose and throat during sinus irrigation.

During follow-up, the necrotic sinus mucosa was eventually replaced by granulation tissue (Figure 5). One month post-injury, the symptoms which bothered the patient most were the residual burning sensation during sinus irrigation and persistent left cheek numbness. Sinus mucosa recovered clinically after three months but cheek numbness persisted. There were no ophthalmic sequelae.

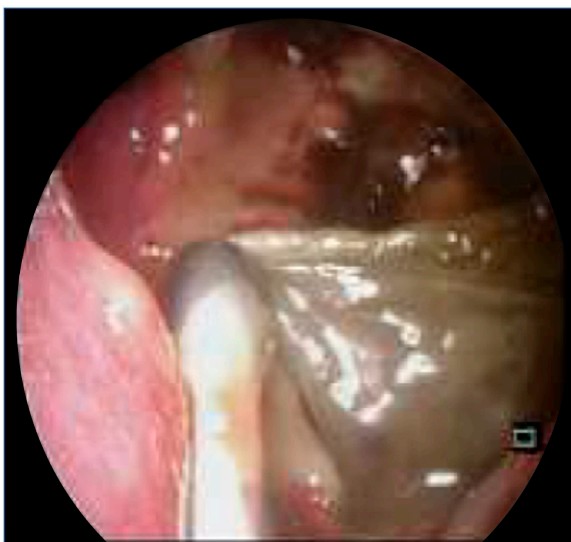

**Figure 4.** Endoscopic clinical image depicting intraoperative findings of denuded left maxillary sinus mucosa, with some remnant dusky mucosa.

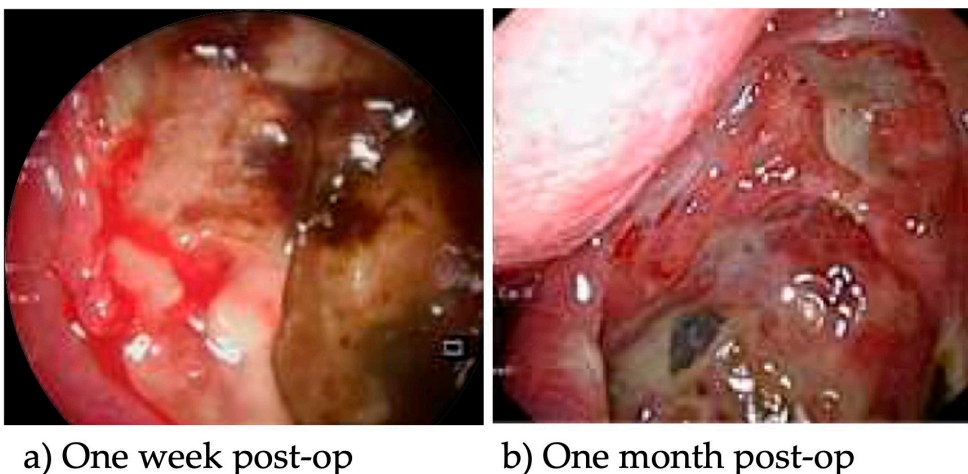

a) One week post-op          b) One month post-op

**Figure 5.** (**a**) Endoscopic clinical image of the left maxillary sinus at one week post-operation, viewed with a 45-degree-angled scope. Remnant dusky mucosa is seen with some slough and granulation; (**b**) Endoscopic clinical image of the left maxillary sinus at one month post-operation, viewed with a 45-degree-angled scope. Mucosa and slough were mostly replaced by granulation.

### 3. Discussion

Sodium hypochlorite, or household bleach, is the most common solution used during root canal treatment to chemically debride the unhealthy tooth root [1,2]. This strongly alkaline solution is used in varying concentrations and induces liquefactive necrosis when it comes into contact with surrounding soft tissue and even bone.

Hypochlorite accidents involve an undesired introduction of sodium hypochlorite into tissues or spaces beyond the root canal, either via periapical extravasation or accidental spillage. Pre-existing anatomical protrusion of the tooth root into the sinus with a dehiscent or thin overlying bony covering predisposed our patient to periapical extravasation. Poor injection techniques are linked to such accidents. Usage of high concentrations of sodium hypochlorite increases the risk and severity of injury [1,2].

Scattered reports of hypochlorite accidents in the existing literature recorded mostly mild and self-limiting superficial soft tissue injury, while severe injuries remain rare. Symptoms usually occur immediately after the accident but may also manifest hours later [2].

All patients experienced acute intense burning pain. Facial and periorbital swelling occurs from extrusion into the subcutaneous tissue. Local bruising, bleeding or hematoma formation can occur. There may be accompanying neurovascular injury, including numbness over the cheek from infraorbital nerve injury, or facial weakness from muscular injury. If injected into the maxillary sinus, patients may experience nasal irritation, discharge or epistaxis [1–3]. Accidental intra-oral spillage can cause intraoral swelling, mucosal ecchymoses and sloughing. There are two reports of airway edema and obstruction [4,5]. There is no existing literature describing the complication of orbital cellulitis. The authors propose that existing dehiscence or chemical erosion of the orbital floor allowed for direct extravasation of the chemical or spread of inflammation into the orbit, resulting in orbital cellulitis in our patient.

The diagnosis of hypochlorite injury is clinical. Chemical sinusitis resulting from hypochlorite accidents can mimic acute complicated bacterial sinusitis. The two may be distinguished via a detailed history and confirmed with thorough examination of the sinuses. Extent of injury and complications can be assessed radiologically.

Principles of management include stabilization of the patient, immediate irrigation to dilute the chemical spillage, anti-inflammatory medications such as non-steroidal anti-inflammatory drugs or steroids for more severe cases, analgesia, and application of external cold compression to the face. Antibiotics are recommended in severe soft tissue injury or superimposed bacterial infection [2,4,5]. Existing guidelines lack recommendations for the management of chemical sinusitis [2].

The authors propose that early sinus surgery be performed, once the diagnosis is entertained, for several reasons. First, for proper examination of the affected sinus to confirm the diagnosis. Second, to facilitate thorough irrigation of the exposed sinus and debridement of devitalized tissue. Third, to obtain tissue for culture to guide subsequent antimicrobial therapy. Finally, to create a wide, patent sinus to monitor the progress of healing and facilitate continued irrigation.

Following the above initial management, close follow-up is indicated to monitor for and treat secondary bacterial sinusitis. Healing of the sinus mucosa is observed via serial endoscopic examination and can take several weeks, as was the case with our patient.

Our patient suffered persistent cheek numbness from infraorbital nerve injury, which bothered him significantly. Unfortunately, prognosis for recovery is guarded, as neuromuscular damage from sodium hypochlorite may be a permanent sequela [2,6].

### 4. Conclusions

Chemical sinusitis and orbital cellulitis are rare but severe and potentially sight-threatening complications of hypochlorite accidents following root canal surgery. Otorhinolaryngologists and Dental surgeons should maintain a high index of suspicion in the correct clinical context to identify and treat the disease expeditiously. A multidisciplinary team of Otorhinolaryngologists, Dental surgeons, Ophthalmologists and Infectious Diseases Physicians are needed to manage this condition effectively. We advocate early surgical intervention once this diagnosis is entertained, in addition to medical therapy. Recovery can take weeks, and the effects of neuromuscular injury may be permanent.

**Author Contributions:** All authors conceptualized the study; T.H.L. and J.J.S. prepared the original draft; All authors reviewed and edited the final manuscript; S.S., V.T. and C.L.N. supervised the study. All authors have read and agreed to the published version of the manuscript.

**Funding:** This research received no external funding.

**Institutional Review Board Statement:** Not applicable.

**Informed Consent Statement:** Informed consent was obtained from all subjects involved in the study.

**Data Availability Statement:** Not applicable.

**Conflicts of Interest:** The authors declare no conflict of interest.

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
