# Peer review of "Bleach-Induced Chemical Sinusitis and Orbital Cellulitis Following Root Canal Treatment"

_2673-351X, doi:10.3390/sinusitis7010002_

Round 1

Reviewer 1 Report

This is an interesting chemical sinusitis case. The study was well designed and structured, however, to improve the quality of manuscript authors need to review for grammar and word choice.

1. Fig 1 & 2 and Fig 5 & 6 could express as single figure.

2.  Line 48 & Fig 3;
   The roots of the second and third maxillary molars.....
  ==> Fig 3 only show the 2nd molar teeth extruded to the sinus cavity.
          please re-check.

3. Line 69;
   Culture-directed antibiotics....
   ==> Bacterial culture usually takes more than a week, but symptoms improved from 24 hours after surgery. Please confirm exactly when antibiotics were used. 

4. Line 58;
 Medial meatal antrostomy 
  ==> middle meatal antrostomy??

Author Response

Dear Reviewer,

Thank you for your valuable suggestions. Please kindly see our point-by-point response below:

1: We have combined Figure 1 and 2, as well as Figure 5 and 6 together as per your suggestion.

2: We have added in another CT scan image cut which shows the dehiscent teeth.  

3: Antibiotics were started immediately after initial clinical assessment, and were changed to culture-directed antibiotics as soon as the bacterial culture and sensitivity results turned positive.   

4: We have edited the typo error. Thank you very much.  

We hope that we have adequately addressed your comments and sincerely thank you for your suggestions.

Reviewer 2 Report

While the term "chemical maxillary sinusitis" is used, it may not be immediately clear to all readers what this refers to. Providing a brief definition or description of the condition in the introduction could help to make the report more accessible.

It is not clear whether the patient had any pre-existing conditions such as cavity problems or gingival infection that may have contributed to their susceptibility to developing chemical maxillary sinusitis.

In Figures 4 and 5, endoscopic clinical image is out of focus. This should be replaced with clear images. In addition, text writings should be utterly removed to hide any personal data.

How long was the patient forced to immersed in the sodium hypochlorite solution. How dense was the concentration and how much was the volume?

The authors reported the complication of orbital cellulitis, however no signs can be detected in the CT figures. This should be described more in detail.

The authors had better describe clinical courses of root canal condition of the upper jaw as well as how to treat them by dental procedures more in detail.

Author Response

Dear Reviewer,

Thank you for your valuable suggestions. Please kindly refer below for our point-by-point response.

Regarding description of chemical maxillary sinusitis:

We have included a brief description of chemical sinusitis in the newly-added introduction section as per your suggestion.

Regarding pre-existing dental conditions:

Based on available medical records, our patient only had a history of dental caries of the left maxillary third molar. Our patient had no other known pre-existing conditions such as cavity problems or gingival infection or any other dental issues.

Regarding endoscopic images:

Unfortunately, based on the images available for download and export, the current image resolution is the best available. While not as ideal, we hope that it is able to portray the pertinent findings we intended to illustrate. We have also removed any text writings as per your suggestion, and ensured that no personal or identification data is shared.

Regarding details of use of hypochlorite solution:

As the patient underwent dental treatment in a separate institution, certain information regarding these questions were not available. Based on our medical records, the concentration of sodium hypochlorite used was 5%. The total volume of sodium hypochlorite used was not charted, but was reported by the dentist to be given in small aliquots of 5ml. The duration of exposure to hypochlorite was also not charted. We have added the relevant information into the manuscript as per your suggestion.

Regarding the complication of orbital cellulitis:

We have added a coronal image depicting fat stranding in the left extra-conal orbital fat. The diagnosis of orbital cellulitis was made based on clinical assessment in conjunction with CT scan findings.

Regarding clinical course of root canal condition:

The treatment of chemical sinusitis secondary to hypochlorite accident would require a multidisciplinary approach. Our case report focuses on the diagnosis, workup and surgical management of this condition from an otolaryngology standpoint with emphasis on the pathology of sinusitis. Details regarding dental treatment and procedures are beyond the intended scope of our discussion.

We hope that we have adequately addressed your comments and sincerely thank you for your suggestions.

Reviewer 3 Report

This is an interesting case report and illustrates the ideal management of maxillary sinus disease as a consequence of contact with sodium hypochlorite solution introduced via root canal treatment. The discussion about potential adverse impact on vision is particularly useful. There is not a large literature on this topic.

Author Response

Dear Reviewer,

Thank you for your encouraging comments. We are grateful to have had the opportunity to contribute to the existing knowledge pool on this rare condition.